# Longitudinal, Intra-Individual Stability of Untargeted Plasma and Cerebrospinal Fluid Metabolites

**DOI:** 10.3390/metabo16010035

**Published:** 2025-12-30

**Authors:** Briana Rocha, Erin M. Jonaitis, Alana Hamwi, Corinne D. Engelman

**Affiliations:** 1Department of Population Health Sciences, School of Medicine and Public Health, University of Wisconsin-Madison, Madison, WI 53726, USA; ahamwi@wisc.edu (A.H.); cengelman@wisc.edu (C.D.E.); 2Wisconsin Alzheimer’s Institute, School of Medicine and Public Health, University of Wisconsin-Madison, Madison, WI 53726, USA; jonaitis@wisc.edu; 3Wisconsin Alzheimer’s Disease Research Center, University of Wisconsin-Madison, Madison, WI 53792, USA

**Keywords:** longitudinal metabolomics, cerebrospinal fluid, plasma, intraclass correlation coefficient, metabolite stability, batch effects, biomarker stability

## Abstract

Background/Objectives: Longitudinal metabolomics analysis offers valuable insights into how metabolic pathways change according to age and health status. However, metabolite levels can fluctuate due to biological factors (e.g., age, diet, and health status) and technical factors (e.g., sample handling, storage times, and instrument performance), with some metabolites exhibiting greater sensitivity to these sources of variability than others. This study aimed to characterize the longitudinal and technical stability of untargeted plasma and cerebrospinal fluid (CSF) metabolites and to identify a subset that remains reliable over the extended time scales required for epidemiological research. Methods: Untargeted ultrahigh-performance liquid chromatography–mass spectrometry (LC-MS) metabolomic profiles were available from multiple visits in the Wisconsin Registry for Alzheimer’s Prevention (WRAP) and Wisconsin Alzheimer’s Disease Research Center (ADRC) studies. For this analysis, we constructed a subset of generally healthy participants with samples drawn at four time points (~2.5 years apart): two visits analyzed in 2017 and two visits analyzed in 2023, corresponding to two distinct analytical waves. We computed Rothery’s intraclass correlation coefficients (ICCs) to quantify intra-wave and inter-wave stability, evaluated pooled quality-control (QC) variation, classified metabolite stability by established thresholds, and developed a composite score integrating longitudinal stability and susceptibility to technical variance. Results: Across all metabolites, median stability was classified as ‘fair’ (Rothery’s *ρ* > 0.40 to ≤0.75) for both plasma and CSF. Although analytical batches were bridged using pooled QC samples, inter-wave stability was significantly lower than intra-wave stability, reflecting increased technical variability across waves. Using the composite score, we identified subsets of metabolites with ‘excellent’ stability and low susceptibility to batch effects in plasma and CSF. Stability patterns varied across biochemical super pathways. Conclusions: This work highlights metabolites suitable for long-term epidemiological studies and informs experimental design and analytical strategies for combining data across cohorts and analytical batches.

## 1. Introduction

Metabolomics has emerged as a powerful tool for identifying predictive and diagnostic biomarkers for a broad range of diseases. Metabolites are functional byproducts of biochemical processes resulting from gene expression, protein interactions, cellular metabolism, and the body’s response to external factors. Untargeted metabolomic studies, which measure thousands of metabolites spanning diverse biochemical pathways, provide a comprehensive snapshot of an organism’s biochemical state, offering valuable insights into health and disease.

Epidemiological studies have successfully used single-time-point metabolomic profiles to identify associations between specific metabolites and a wide range of diseases, including Alzheimer’s disease (AD) [1,2], cardiovascular disease [3], and cancer [4]. The growing availability of large-scale, longitudinal metabolomic data presents new opportunities to explore causal relationships between metabolites and disease progression. However, identifying meaningful metabolic changes over extended periods is challenging, especially for metabolites with high variation.

Metabolite variation may occur due to several biological and technical sources. Biological variation can arise from exogenous factors such as diet, medication usage, and environmental exposures, as well as endogenous factors like circadian rhythms and hormonal cycles [5]. Metabolite levels may also shift with age [6,7,8], health status [9], and lifestyle [10], further contributing to biological variability. In contrast, technical variability can stem from sample handling, long-term storage, freeze–thaw cycles, and instrumentation drift, all of which may result in batch effects that complicate downstream analyses [11]. Longitudinal studies, which often span many years and multiple waves, are particularly susceptible to both biological and technical sources of variation. Moreover, to achieve sufficient sample sizes, data from multiple sites or cohorts are frequently combined, introducing additional heterogeneity [12].

Although several studies have examined the reliability of longitudinal metabolite measurements, the number of time points has been limited, most have focused on select, clinically relevant metabolites [13,14,15], and none have characterized the reliability of untargeted metabolites detected in cerebrospinal fluid (CSF). For neurodegenerative diseases such as Alzheimer’s disease (AD), CSF metabolites are particularly relevant, as CSF is in direct contact with the brain, where biochemical processes associated with neuropathology and neuroinflammation typically occur years before the onset of clinical symptoms. Consistent with this, associations between CSF metabolites and clinical AD status, the presence of pathological amyloid and tau, AD-related markers of neuroinflammation and cognitive status have been reported [16,17,18,19]. As such, characterizing the longitudinal reliability of CSF metabolites represents a critical step toward identifying metabolic changes that coincide with AD progression.

Most prior studies of longitudinal metabolite stability used intraclass correlation coefficients (ICCs) to classify and identify a subset of plasma metabolites that were ‘excellently’ stable over the intervals tested [20,21,22,23,24]. However, standard ICC analysis assumes normality, which is often violated in metabolomics data due to skewed, zero-inflated, or censored distributions. Traditional approaches to meet this assumption, such as log transformations, do not always achieve normality and can obscure biologically relevant outliers, further complicating efforts to detect disease-related metabolic shifts. An alternative is to use non-parametric methods, such as Rothery’s non-parametric ICC [25], which can handle non-normal distributions and is suitable for the diverse metabolomic profiles encountered in large, untargeted metabolomic data sets.

In the present study, we apply Rothery’s non-parametric ICC to evaluate intra- and inter-wave stability of plasma and CSF metabolites measured at four time points collected approximately 2.5 years apart. The data were drawn from generally healthy participants enrolled in two longitudinal cohorts: the Wisconsin Registry for Alzheimer’s Prevention (WRAP) and the Wisconsin Alzheimer’s Disease Research Center (ADRC). For our analysis, metabolomic measurements were conducted in 2017 (using samples from visits 1 and 2) and 2023 (visits 3 and 4). To reduce batch effects, a pool of approximately 300 representative participant samples from the first wave was used to bridge the two batches. To assess both temporal and technical variation, we examined multiple pairwise comparisons: intra-wave stability (e.g., visits 1 and 2, and visits 3 and 4), as well as inter-wave stability (visits 2 and 3), where samples were processed roughly 6 years apart but collected an average of 2.5 years apart.

We classified metabolites based on their stability, examined stability patterns by super pathway, and conducted comparative analyses to assess differences in stability between plasma and CSF for metabolites detected in both fluids. To further evaluate technical reliability, we calculated coefficients of variation (CVs) using pooled quality control (QC) samples and estimated both intra- and inter-wave Rothery ICCs. This allowed us to identify metabolites highly susceptible to technical variation and batch effects. Based on these metrics, we developed a composite stability score to highlight metabolites that are both longitudinally stable and robust to sources of technical variation.

By characterizing the longitudinal stability of a broad range of plasma and CSF metabolites, our findings provide guidance for both study design and post hoc interpretation in longitudinal metabolomics studies, particularly in scenarios that require combining data across cohorts, sites, and/or analytical batches.

## 2. Materials and Methods

### 2.1. Study Participants

Longitudinal human plasma and CSF samples were obtained from participants enrolled in WRAP and the Wisconsin ADRC studies. The same collection protocols were used for both studies (see Plasma and CSF Collection sections). WRAP began recruitment in 2001 as a prospective cohort study, with initial follow-up 4 years after baseline and subsequent, ongoing follow-up every 2 years. WRAP enrolled middle-aged adults (between 40 and 65 years) enriched for a parental history of AD who were non-demented at baseline [26]. Participant visits took place at one of three sites in Wisconsin: Madison, Milwaukee, and La Crosse. The Wisconsin ADRC began recruitment in 2009, enrolling healthy controls as well as individuals with mild cognitive impairment (MCI), dementia, or clinically diagnosed AD. For cognitively unimpaired individuals younger than 65 years, follow-up visits occurred every 2 years. Once participants reached age 65 and/or became cognitively impaired, visit frequency increased to once per year. All visits took place in Madison, Wisconsin [27]. For the purposes of this study, individuals with diagnosed AD, dementia, or AD-related pathophysiology were excluded (see methods sections on Data Preparation and Exclusion of Individuals with AD-related Pathology).

### 2.2. Plasma Collection

Participants underwent venipuncture, most after fasting for ≥8 h, and provided 30 mL of blood into 3 × 10 mL lavender top EDTA tubes (BD 366643; Franklin Lakes, NJ, USA). Samples were mixed gently by inverting 10–12 times and were centrifuged for 15 min at 2000× *g* at room temperature within 1 h of collection. Plasma samples were aliquoted into 2 mL cryovials (Wheaton Cryoelite W985863; Millville, NJ, USA). Aliquoted plasma was frozen at −80 °C within 90 min and stored until overnight shipment to Metabolon, Inc. (Durham, NC, USA), which similarly kept samples frozen at −80 °C until analysis. For this study, participants who fasted <8 h prior to plasma collection were excluded.

### 2.3. CSF Collection

Participants underwent lumbar puncture (LP) and provided 22 mL of CSF via gentle extraction with a Sprotte 24- or 25-gauge needle. After collection, samples were transferred into a single 30 mL polypropylene tube, gently mixed, and centrifuged for 10 min at 2000× *g* at 4 °C within 30 min of collection, before aliquoting in 0.5 mL portions for freezing and storing at −80 °C. Beginning in December 2018, CSF storage protocols were standardized. Accordingly, all CSF samples were aliquoted and frozen in clear, 0.5 mL Sarstedt cryovials. Prior to this, the storage tube type varied by clinical site. Samples were kept frozen until they were shipped overnight to Metabolon, Inc. (Durham, NC, USA), which similarly kept samples frozen at −80° C until analysis.

### 2.4. Thresholds for Exclusion of Individuals with AD-Related Pathology

#### 2.4.1. Biomarker Detection and Thresholds

Each biofluid sample was assigned a binary ‘positive’ or ‘negative’ value for amyloid and tau. The details of the threshold-setting process are described elsewhere [28]. Briefly, for CSF amyloid positivity, receiver-operator characteristic (ROC) curves were fit comparing the Aβ42/Aβ40 to the amyloid PET gold standard, and the threshold was selected that maximized Youden’s index. For CSF tau positivity, no gold standard was available, and so the threshold was set at the upper side of a nonparametric 95% confidence interval around the mean pTau181 result in a group of young (40–60 years), amyloid-negative, cognitively unimpaired participants. CSF samples positive for both amyloid and tau were excluded from the present analysis.

For plasma samples, pTau217 was used to determine AD biomarker positivity since it has been shown to correlate with amyloid PET and represents tau proteinopathy in response to amyloid plaques [29]. Therefore, the pTau217 level is an indicator of both amyloid and tau positivity. Ashton et al.’s high-specificity cutoff of >0.63 pg/mL was used to identify and exclude samples likely to be both amyloid- and tau-positive [30].

#### 2.4.2. Exclusion Due to Clinical Diagnosis of Dementia

Cognitive status was determined via a multidisciplinary clinical consensus conference. The process has been described in detail elsewhere [26]. Briefly, a panel of geriatricians, neuropsychologists, and nurse practitioners examines the cognitive performance of Wisconsin ADRC and WRAP participants and assigns a cognitive status based on National Institute on Aging–Alzheimer’s Association (NIA–AA) criteria [29,31]. Individuals with dementia were excluded from the analysis.

### 2.5. Untargeted Metabolomics

Plasma and CSF metabolomics were performed in two waves (Figure 1). The first wave included CSF and plasma samples collected between 2010 and 2017. The second wave of samples was collected between 2017 and 2023. For each wave, samples were pulled by the Wisconsin ADRC and WRAP Biospecimen Laboratory and sent to Metabolon, Inc. (Durham, NC, USA) for analysis using their global untargeted metabolomics platform.

Before extraction, several recovery standards were added for QC purposes. Proteins were precipitated with methanol under vigorous shaking for 2 min (Glen Mills GenoGrinder 2000, Glen Mills, Clifton, NJ, USA), followed by centrifugation. The resulting extract was divided into four fractions for analysis: two separate reverse phase (RP)/UPLC-MS/MS methods with positive-ion mode electrospray ionization (ESI), one for analysis by RP/UPLC-MS/MS with negative-ion mode ESI, and one for analysis by HILIC/UPLC-MS/MS with negative-ion mode ESI. Samples were placed briefly on a TurboVap^®®^ (Zymark, Hopkinton, MA, USA) to remove the organic solvent before storage overnight under nitrogen.

Several types of controls were analyzed in concert with the experimental samples: A pooled matrix sample generated by taking a small volume collected from 300 representative experimental samples served as a technical replicate throughout the data set; extracted water samples served as process blanks; and a cocktail of QC standards that were carefully chosen not to interfere with the measurement of endogenous compounds were spiked into every analyzed sample. Instrument variability was determined by calculating the median relative standard deviation (RSD) for the standards that were added to each sample prior to injection into the mass spectrometers. Experimental samples were randomized across the platform run with QC samples spaced evenly among the injections.

Metabolon used a ACQUITY ultra-performance liquid chromatography (UPLC; Waters, Millford, MA, USA) and a Q-Exactive high-resolution/accurate mass spectrometer (Thermo Scientific, San Jose, CA, USA) interfaced with a heated electrospray ionization (HESI-II) source and Orbitrap mass analyzer operated at 35,000 mass resolution (PMID: 32445384).

Dried sample extracts were reconstituted in solvents compatible with each of the four analysis methods. One aliquot was analyzed using acidic positive ion conditions, chromatographically optimized for more hydrophilic compounds. In this method, the extract was gradient-eluted from a C18 column (Waters UPLC BEH C18-2.1 × 100 mm, 1.7 µm, Millford, MA, USA) using water and methanol, containing 0.05% perfluoropentanoic acid (PFPA) and 0.1% formic acid (FA). Another aliquot was also analyzed using acidic positive ion conditions; however, it was chromatographically optimized for more hydrophobic compounds. In this method, the extract was gradient-eluted from the same aforementioned C18 column using methanol, acetonitrile, water, 0.05% PFPA, and 0.01% FA and was operated at an overall higher organic content. Another aliquot was analyzed using basic negative-ion optimized conditions using a separate dedicated C18 column. The basic extracts were gradient-eluted from the column using methanol and water, however, with 6.5 mM Ammonium Bicarbonate at pH 8. The fourth aliquot was analyzed via negative ionization following elution from a HILIC column (Waters UPLC BEH Amide 2.1 × 150 mm, 1.7 µm, Millford, MA, USA) using a gradient consisting of water and acetonitrile with 10 mM Ammonium Formate, pH 10.8 (HILIC). Raw data were extracted, peak-identified, and QC-processed using a combination of proprietary, Metabolon-developed software.

### 2.6. Data Bridging and QC

Following the second wave of metabolomics, Metabolon provided bridged data sets that included batch-normalized measurements from both waves. Only metabolites that were detected in both waves were included in the bridged data sets. Metabolite measurements from the pooled QC samples collected in 2017 were used to bridge the 2017 and 2023 data sets and remove batch effects from both participant and QC samples. Metabolite measurements were scaled to the median measurement for that metabolite as measured across all pooled QC samples.

Samples with missing values for more than 40% of metabolites were removed (0 plasma samples, 2 CSF samples) prior to analysis. Metabolites with missing values for >75% of all samples were also removed (51 metabolites from CSF and 182 metabolites from plasma). For each participant and pooled QC sample, a local outlier factor (LOF) was calculated using measurements from all metabolites with <20% missingness [32].

Histograms of LOFs were plotted and used to determine thresholds for flagging sample outliers in plasma and CSF data sets (Appendix A). Nine plasma and four CSF samples were identified and removed (Appendix A). Although the bridged QC samples were generally represented in the smaller peak (lower LOFs), some QC samples exceeded the identified threshold (dashed line in Appendix A) and were excluded (4 CSF QC samples).

### 2.7. Data Filtering

Figure 2 displays the order in which filters were applied, and the number of samples and individuals remaining after each filter was implemented. As described above, nine plasma and six CSF participant samples were removed in the QC step. Plasma samples from individuals who fasted less than 8 h prior to venipuncture or who did not provide fasting status were removed (*n* = 294 samples). Fasting status was not considered for CSF samples as participants were not required to fast prior to lumbar puncture appointments, and our own analysis, as well as work from others [33], has shown that CSF metabolites are minimally affected by fasting status. Our analysis revealed only two CSF metabolites that were associated with fasting status in CSF: the lipid 3-hydroxybutyrate (BHBA) and the nucleotide orotate.

Samples taken after an individual received a clinical diagnosis of dementia were excluded from this study (see Clinical Diagnoses methods section). Additionally, samples collected at the same time as, or after an individual tested positive for AD-related biomarkers, were excluded from this study (see AD Biomarker Detection methods section). As a result, 629 samples were removed from the plasma data set, and 411 samples were removed from the CSF data set.

To ensure a generally healthy population of participants, individuals who self-reported major cardiovascular disease (including a history of heart attack(s), coronary bypass surgery, stent or pacemaker placement, or congestive heart failure) or diabetes (Type I or II) were excluded; 658 plasma samples from 262 individuals, and 142 CSF samples from 92 individuals were removed for reporting one or more of these comorbidities.

After the above filters were applied, remaining observations from the same individual that were greater than 6 years apart were removed (5 samples from plasma and 27 from CSF).

### 2.8. Compiling Data Sets

We compiled data sets consisting of individuals who, after QC and filtering, had ≥2 samples with metabolomics data from wave 1 (2017) and ≥2 samples from wave 2 (2023). The final data sets consisted of 278 plasma donors and 33 CSF donors. For each individual, 4 samples were selected. To minimize age differences between samples, when an individual had >2 samples for a given wave, the two most recent samples from wave 1 and the two earliest samples from wave 2 were selected. After compiling the data sets, visit numbers 1–4 were assigned de novo, with visits 1 and 2 belonging to wave 1 and visits 3 and 4 belonging to wave 2. The inter-wave data set contained visit 2 from wave 1 and visit 3 from wave 2.

### 2.9. Statistical Analyses

#### 2.9.1. Intraclass Correlation

Intraclass correlation is commonly used to report metabolite variability. Traditional ICC assumes normality, whereas Rothery’s ICC can be used for non-parametric distributions. To test the consistency of traditional and Rothery’s ICC, we calculated both metrics on a subset of normally distributed metabolites. To identify these, we calculated skewness and performed a Kolmogorov–Smirnov (KS) test for each metabolite distribution in each unique data set (wave 1, inter-wave, and wave 2). Metabolites with approximately normal distributions (KS *p*-value >0.05 and skewness <1) are listed in Appendix A. To ensure adequate sample sizes and increase power, only metabolites for which >20 participants had complete, unimputed measurements were analyzed.

Traditional intraclass correlations were calculated using the irr package in R. A two-way mixed, single-rater, consistency model was specified. We then computed a nonparametric measure of intraclass correlation following Rothery (1979) [25]. This coefficient reflects how strongly measurements cluster within individuals over time, based on their rank order. Specifically, it estimates the probability that two measurements from the same individual are more similar to each other (in rank) than values from different individuals. In other words, as subject ID imposes clustering by rank, Rothery’s intraclass correlation coefficient *ρ*_c_ will increase. On its native scale, this coefficient is bounded by the interval [2/3, 1]. However, Rothery derived a monotonic transformation to the approximate interval [0, 1] to better map to users’ intuitions about intraclass correlation coefficients. Like traditional ICC, it is mathematically possible for this transformed coefficient (*ρ*) to be negative. Such cases arise when within-individual measurements are less rank-consistent than expected under the null hypothesis that no within-individual clustering exists. We created an R module to compute *ρ* as described in the original paper (see source code in Supplement). We used the method of normal approximation to calculate a confidence interval for the coefficient on its native scale; these confidence limits were then transformed using the same monotonic equation applied to *ρ*_c_. Using these point estimates and confidence limits, we examined the intra- (*ρ*_1_ and *ρ*_2_) and inter- (*ρ* _1:2_) wave stability of each metabolite in plasma and CSF. The same established thresholds for rating traditional ICC values were applied. Metabolites with Rothery’s *ρ* >0.75 were rated as ‘excellent’, >0.40 and ≤0.75 were rated as ‘fair’, and ≤0.40 were rated as ‘poor’. To confirm that Rothery’s method performs similarly to the classical ICC when normality assumptions are met, we calculated both Pearson’s and Spearman’s correlation coefficients on the subset of normally distributed metabolites.

#### 2.9.2. Coefficient of Variation of QC Samples

For all analyses, bridged QC samples were used (see Data Bridging and QC section). Given that the QC samples were aliquoted from the same pool of participant samples, variation in these measurements within and across analytical runs represents sources of technical variation. For each data set (wave 1, inter-wave, and wave 2), the mean and standard deviation for each metabolite (normally and non-normally distributed) were calculated for all pooled QC samples that were run alongside the included participant samples. The coefficient of variation (CV) for QC samples (*CV_QC_*) was calculated by dividing the standard deviation (SD) by the mean and multiplying by 100.

#### 2.9.3. Composite Stability Score

We developed a composite score that integrates multiple measures of reliability. For each metabolite (normally and non-normally distributed), the composite score was calculated as the average of the mean intra-wave *ρ* values (*ρ*_1_, *ρ*_2_), and inter-wave (*ρ*_1:2_) *ρ* value, which together reflect the combination of biological and technical variation, minus the average of the mean normalized intra-wave *CV_QC_* value (${\hat{CV}}_{\mathrm{QC}1}$, and ${\hat{CV}}_{\mathrm{QC}2}$), and inter-wave *CV_QC_* value (${\hat{CV}}_{\mathrm{QC}1:2}$) which capture technical variation.

All CV_QC_ values were normalized (${\hat{CV}}_{\mathrm{QC}}$) to achieve a range of 0–1, where 0 represents the minimum observed *CV* value and 1 represents the maximum observed *CV* value across all participants and *QC* samples. The formula gives equal weight to intra- and inter-wave stability while accounting for technical variance:$$\frac{\left(\frac{\rho_{1}+\rho_{2}}{2}\right)+\rho_{1:2}}{2}\;-\frac{\left(\frac{\hat{{CV}_{QC1}}+\hat{{CV}_{QC2}}}{2}\right)+\hat{{CV}_{QC1:2}}}{2}$$

By including distinct terms for inter-batch stability (*ρ*_1:2_) and variance (${\hat{CV}}_{\mathrm{QC}1:2}$), this metric places greater emphasis on cross-wave stability than traditional ICC metrics and penalizes metabolites with high technical variance. Like *ρ*, the composite score has a maximum value of 1, which is achieved when all *ρ* values are 1, and all normalized *CV_QC_* (${\hat{CV}}_{\mathrm{QC}}$) values equal 0. Negative composite scores are possible when the averaged ${\hat{CV}}_{\mathrm{QC}}$ values exceed the averaged *ρ* values.

## 3. Results

### 3.1. Summary of Participants

The baseline characteristics of participants enrolled in the WRAP or Wisconsin ADRC studies with four sequential visits are shown in Table 1. There were 1112 plasma samples from 278 individuals and 132 CSF samples from 33 individuals available for analysis. For plasma, the mean age at baseline was 61.2 for wave 1 (visits 1 and 2), 63.5 for the inter-wave subset (visits 2 and 3), and 66 for wave 2 (visits 3 and 4). For CSF, the mean age at baseline was 62.2 for wave 1, 65 for the inter-wave sample, and 67.4 for wave 2. The average time interval between visits 1 and 2 was 2.4 years for wave 1, 2.5 years for the inter-wave sample, and 2.9 years for wave 2 for the plasma samples. The average time interval between visits 1 and 2 was 2.7 years for wave 1, 2.4 years for the inter-wave sample, and 2.3 years for wave 2 for the CSF samples: 68.4% of plasma samples and 63.6% of CSF samples were from females; 100% of the plasma samples and 42.4% of the CSF samples were from WRAP (the plasma samples in the Wisconsin ADRC were initially collected in heparin tubes and were not sent to Metabolon, so no Wisconsin ADRC participants had four sequential plasma samples).

### 3.2. ICC and Rothery’s ρ Are Highly Correlated for Normally Distributed Metabolites

To test the correlation between traditional ICC and Rothery’s non-parametric ICC, we calculated both metrics on a subset of metabolites with approximately normal distributions (KS-test *p*-value < 0.05 and skewness < 1). When a metabolite met these criteria in only a subset of datasets, calculations were limited to the approximately normal datasets. Overall, 173 unique plasma and 227 unique CSF metabolites showed approximate normality in at least one data set (Appendix A; total observations = 886). Since Rothery’s ICC is a rank-based test, we calculated Spearman’s rank correlation coefficient (Spearman’s r) as well as Pearson’s correlation coefficient (Pearson’s r). For the subset of normally distributed metabolites, these metrics are highly correlated (Spearman’s r: 0.94; Pearson’s r: 0.93; Figure 3a). Rothery’s ICC is more robust to deviations from normality, which can attenuate stability estimates in traditional ICC. Accordingly, including all metabolites regardless of distribution weakens the correlation (Spearman’s r: 0.86; Pearson’s r: 0.85; Figure 3b).

### 3.3. Stability Across All Metabolites in Plasma and CSF

To evaluate the stability of metabolites in plasma and CSF, we applied Rothery’s non-parametric ICC (*ρ*) to all metabolites with complete measurements in at least 20 participants. Across all datasets, median *ρ* values indicated moderate stability (range: 0.50 to 0.70), with the majority of metabolites given a ‘fair’ rating (range: 57.2–71.7%; Figure 4a, Table 2). Pairwise comparisons of *ρ* values were performed using two-sided Wilcoxon rank-sum tests. In both plasma and CSF, inter-wave stability was significantly lower than intra-wave stability (*p* <0.001). Across all plasma metabolites, *ρ* values were comparable between wave 1 and wave 2, whereas CSF metabolites showed a significant decline in stability in wave 2.

To directly compare metabolite stability between plasma and CSF, we focused on the 265 metabolites detected in both fluids. Across all datasets, median *ρ* values for these shared metabolites were significantly higher in CSF compared to plasma (Figure 4b), suggesting generally greater metabolite stability in CSF.

### 3.4. Stability by Pathway

We next assessed metabolite stability at the super pathway level. Xenobiotics, which are derived from exogenous sources, as well as ‘unnamed’ and ‘partially characterized molecules’ for which super pathway information is unavailable, were excluded from this analysis, leaving 689 plasma and 238 CSF metabolites. Similar to what was observed across all metabolites, at the super pathway level, median stability was generally lower for inter-wave sample pairs compared to intra-wave pairs for both plasma and CSF (Appendix A; Appendix A). Among the super pathways, only amino acids showed significant differences in *ρ* values between wave 1 and wave 2, with stability decreasing in wave 2 for both plasma (*n* = 189 metabolites) and CSF (*n* = 118 metabolites). Amino acids represent the largest super pathway in CSF, suggesting that this group may be a key contributor to the overall decline in stability observed in wave 2 (Figure 4a).

In plasma, the cofactors and vitamins super pathway (*n* = 30 metabolites) demonstrated the highest median stability rating (*ρ* = 0.63 for waves 1 and 2, and inter-wave *ρ* = 0.55). In CSF, despite having the highest *CV_QC_* values, the lipid super pathway (*n* = 52 metabolites) had the highest intra- and inter-wave stability ratings (wave 1 *ρ* = 0.76; wave 2 *ρ* = 0.79; inter-wave *ρ* = 0.69).

### 3.5. Composite Score Offers More Nuanced Assessment of Longitudinal Metabolite Stability

Both traditional and Rothery’s ICC capture the combined effects of biological and technical variation and, when computed across all samples, inherently include intra-batch stability. However, longitudinal studies often span multiple analytical waves, and the ideal metabolites for such designs are those that remain highly stable over time and are robust to batch effects. To identify these, we developed a composite stability score that integrates intra-wave reproducibility (the average of *ρ*_1_ and *ρ*_2_) and inter-wave reproducibility (*ρ*_1:2_) and applies a penalty for high technical variance (${\hat{CV}}_{\mathrm{QC}}$). The resulting score ranges from –0.23 (worst; the xenobiotic benzoate in CSF) to 0.95 (best; the amino acid N6-methyllysine in CSF), with a theoretical maximum of 1.0, which can be achieved when all intra- and inter-wave *ρ* values equal 1.0 and all ${\hat{CV}}_{\mathrm{QC}}$ values are zero.

Figure 5 shows composite scores across all metabolites in plasma (a and c) and CSF (b and d). The *x*-axis represents the average intra-wave stability (*ρ*_1_ and *ρ*_2_), and the *y*-axis represents inter-wave stability (*ρ*_1:2_). As expected, the highest scores (yellow points) cluster in the upper-right quadrant, showing strong intra- and inter-wave stability and minimal technical noise (point size; Figure 5c,d). In contrast, low-scoring metabolites generally appear in the lower left quadrant, reflecting ‘poor’ overall stability. Importantly, for CSF, the upper right quadrant also contains several metabolites from the lipid super pathway that, despite displaying ‘excellent’ intra- and inter-wave stability, exhibit high levels of technical variability resulting in poor to moderate composite scores (Figure 5d; dashed pink boxes; highlighted in Table 3). Additionally, a number of metabolites appear in the lower middle quadrants for both plasma and CSF (Figure 5c,d; solid pink boxes; Table 3), where intra-wave stability is ‘fair’, but inter-wave reproducibility and/or technical variance are/is poor. Notably, CSF glutamate, which displayed ‘fair’ average intra-wave stability (*ρ* = 0.52), was among the lowest-scoring metabolites due to ‘poor’ inter-wave stability (*ρ* = −0.45). These cases illustrate how ICC metrics alone may overestimate stability, whereas the composite score penalizes such discrepancies, providing a more nuanced assessment of reliability.

In plasma, top-scoring metabolites exhibited ‘excellent’ intra- and inter-wave stability (*ρ* ≥ 0.93 and *ρ* ≥ 0.88, respectively) and relatively low normalized technical variation (${\hat{CV}}_{\mathrm{QC}}$ ≤ 0.03). These included the amino acid metabolites N2-acetyl,N6-methyllysine, N6,N6-dimethyllysine, and N2-acetyl,N6,N6-dimethyllysine and the lipids 5alpha-androstan-3alpha,17beta-diol monosulfate (2), and androstenediol (3alpha, 17alpha) monosulfate (3). The lowest-scoring plasma metabolites had composite scores ranging from 0.0 to 0.08 and included the lipid metabolite chiro-inositol, the carbohydrate metabolite maltotriose, the nucleotide cytosine, and the xenobiotic compounds, benzoate, and iminodiacetate (IDA). For these metabolites, the low composite scores were driven by both ‘poor’ average intra-wave stability (*ρ* values between 0.18 and 0.21) and ‘poor’ inter-wave stability (*ρ* values between –0.08 and 0.06).

A similar pattern was observed in CSF. The top-scoring CSF metabolites included the amino acid metabolites N6-methyllysine, ethylmalonate, homocarnosine, and N-acetylputrescine, as well as the partially characterized molecule 2-amino-4-cyanobutanoate. All showed consistently high intra- and inter-wave stability (*ρ* ≥ 0.87 and *ρ* ≥ 0.84, respectively) and low technical variability (${\hat{CV}}_{\mathrm{QC}}$ ≤ 0.02), yielding composite scores between 0.86 and 0.95. Meanwhile, the lowest-scoring CSF metabolites included the amino acid glutamate, the energy-related compound isocitrate, and the xenobiotic compounds 4-chlorobenzoic acid, tartarate, and benzoate. The lowest-scoring metabolites had scores ranging from −0.23 to 0.06. Notably, glutamate had ‘fair’ average intra-wave stability (*ρ* = 0.41) but a sharply negative inter-wave stability (*ρ* = −0.45), leading to a composite score of 0.03 despite minimal QC variability (highlighted in Figure 5d). This again demonstrates how the composite score effectively penalizes metabolites that appear stable in a limited context but are inconsistent over time or across batches. Composite scores and stability metrics for all metabolites can be found in Appendix A.

### 3.6. Composite Score Reveals No Significant Difference in Stability Between Plasma and CSF Metabolites

Given that the composite score presents a more nuanced assessment of longitudinal metabolite stability, we compared composite scores for the 265 metabolites detected in both plasma and CSF. When comparing Rothery’s *ρ* values, CSF metabolites were significantly more stable than plasma metabolites across all datasets (Figure 4b). However, we found no significant difference in composite scores between plasma and CSF (Figure 6). Across all data sets, CSF metabolites displayed significantly higher susceptibility to technical variation (*CV_QC_*) than plasma (Appendix A), suggesting that the penalty applied for technical variation is driving composite scores down for CSF metabolites.

## 4. Discussion

As large-scale metabolomic datasets become increasingly available, there is growing interest in leveraging longitudinal data to move beyond cross-sectional associations toward identifying causal metabolic pathways underlying disease processes. Such studies often require sampling over extended time periods and integrating data across cohorts, study designs, and analytical batches. While necessary, these practices introduce biological and technical variability that can obscure true disease-related signals.

Although several studies have reported on the longitudinal stability of metabolites, few have explicitly examined cross-batch stability, and data on the temporal reliability of CSF metabolites remain limited. This study addresses these gaps by characterizing the stability of untargeted plasma and CSF metabolites in healthy, cognitively unimpaired individuals across four visits spanning approximately 6 years. Using Rothery’s non-parametric intraclass correlation coefficient (ICC), we assessed both intra- and inter-wave reliability. In addition, we developed a composite score that integrates these reliability measures with technical variance to identify metabolites that are temporally stable, analytically reproducible, and resilient to batch effects, critical features for robust longitudinal epidemiological analyses. Importantly, this composite score is readily applicable to other longitudinal or multi-batch metabolomic datasets where it can be used retrospectively to evaluate the reliability of key findings or prospectively to prioritize stable metabolites, thereby supporting more informed study design and interpretation.

Rothery’s non-parametric ICC offered key advantages for untargeted metabolomics, particularly in epidemiological settings where outliers may reflect meaningful biological variation. Unlike traditional parametric ICC, it does not assume normality, making it well-suited for the skewed and/or left-censored distributions often observed in metabolomics data. This allowed us to analyze untransformed data. Moreover, Rothery’s monotonic scaling transformation preserves interpretability on the same scale as traditional ICC, enabling the use of established thresholds for classifying stability.

Rather than calculate ICC across all visits combined, we assessed stability within each analytical wave separately. This approach allowed us to compare stability between waves. While intra-wave metabolite stability was generally consistent across waves in plasma, we observed a significant decline in CSF stability for wave 2 (visits 3–4; Figure 4a). At the super pathway level, only amino acid-related metabolites showed significantly reduced stability in wave 2 (Appendix A; Appendix A). Because amino acids represent the largest super-pathway in CSF, this pattern suggests that they may be driving the overall decrease in CSF stability between waves. Other pathways exhibited modest increases in median stability in wave 2, but these differences were not statistically significant. As age [6,7,8], health status [7,9], and lifestyle factors [34,35] have all been associated with metabolic changes, future studies should investigate whether these influence metabolite stability.

To more directly assess susceptibility to batch effects, we calculated inter-wave stability (visits 2–3). Although the use of pooled quality-control (QC) samples is widely considered the gold standard for cross-run normalization, our findings show that they do not fully eliminate batch-related variability, particularly for sensitive metabolites. Inter-wave stability was significantly lower than intra-wave stability for both plasma and CSF. However, the degree of susceptibility varied across metabolites. Some showed negligible differences between intra- and inter-wave stability, while others, such as CSF glutamate, differed by as much as 97% (Figure 5; Table 3). Despite this extreme sensitivity to batch effects, the *ρ* value for glutamate still falls within the ‘fairly stable’ range (0.4–0.75); whereas its composite score, which gives equal weight to intra- and inter-wave stability, ranks it among the least reliable CSF metabolites. In contrast, plasma glutamate only showed a 1% difference between intra- and inter-wave stability (Appendix A), suggesting it may be more suitable for longitudinal analysis in our dataset.

The relatively high *CV_QC_* values seen in CSF affected composite scores. Several CSF metabolites with ‘excellent’ intra- and inter-wave ICC values (ICC > 0.75) received only moderate composite scores due to elevated technical variance (Figure 5d, pink dashed box). As a result, although CSF metabolites had significantly higher median Rothery’s ICC values than plasma, composite scores were not significantly different between fluids. These findings illustrate how reliance on ICC alone may obscure important susceptibilities to technical variance, whereas the composite score provides a more nuanced and stringent measure of metabolite reliability for longitudinal analyses.

In both fluids, metabolites involved in amino acid metabolism were among the highest-scoring compounds, accounting for three of five top-ranked metabolites in plasma and four of five in CSF (Table 3, Appendix A). Given that amino acids are physiologically ubiquitous and subject to tight homeostatic regulation, their representation among the highest-scoring metabolites is unsurprising. However, as evidenced by the batch sensitivity observed for glutamate, as a class, amino acids are not inherently resistant to batch effects. Although two androgenic steroids from the lipid super pathway were among the top-ranked metabolites in plasma, lipids were notably absent from high-ranking CSF metabolites. This divergence is likely due to the elevated CV values observed in CSF lipids and underscores important pathway-specific differences in susceptibility to sources of technical variance.

Previous studies have suggested that CSF metabolites are more stable than those in plasma, likely due to reduced sensitivity to short-term influences such as diet [34] and lower technical variance [36]. Whereas Crews et al. reported lower median CVs in CSF relative to plasma, we observed the opposite (Appendix A). This discrepancy likely reflects differences in study design and sources of technical variation. Crews et al. assessed variability across individuals using biological replicates. In contrast, we focused on technical reproducibility across batches using pooled QC samples. Moreover, CSF samples in our study were stored in multiple tube types prior to 2018, when CSF storage protocols were standardized across sites. As the precise sources of technical variation and the extent to which they affect variability cannot be fully known, making direct comparisons between studies is difficult.

In our data, elevated technical variance in CSF was driven primarily by metabolites in the lipid super-pathway, which displayed markedly higher *CV_QC_* values in CSF than in plasma (Appendix A). As lipids are known to be particularly sensitive to pre-analytical conditions [37], the lack of standardized CSF tube types prior to 2018 therefore represents a plausible contributor to the elevated variability observed in lipids in earlier analytical waves. While retrospective harmonization is often unavoidable in long-running cohort studies, awareness of these sources of variability is essential.

Our results should be interpreted with several limitations in mind. First, ICC values, whether parametric or non-parametric, capture technical variance, which is influenced by study design, sample handling, and platform. While our findings align with prior work using similar methods [21], our results should not be interpreted as a quantitative assessment of inherent metabolite stability. Rather, they reflect overall patterns of reliability and susceptibility to batch effects within the context of our study design. As such, findings may not generalize to other studies, particularly when protocols and platforms differ greatly. Second, we excluded individuals with missing metabolite values rather than imputing them. While this avoids assumptions inherent to imputation, it may slightly inflate stability estimates, as metabolites with lower stability tend to have more missing values. Third, sampling intervals were not uniform across participants. Although extreme outliers were removed, this variability may have introduced additional noise. Finally, the retrospective nature of the study design limited our ability to control pre-analytical factors, including changes in CSF storage protocols over time. Whenever possible, care should be taken to standardize pre-analytical protocols for longitudinal metabolomics studies, particularly when sensitive molecular classes such as lipids are of interest. In cases such as ours, when retrospective study design is unavoidable, incorporating technical variance into reliability assessments is critical.

Despite these limitations, this study provides a valuable framework for assessing metabolite reliability using both biological and technical measures of variation. Our composite score offers a more stringent and informative metric than ICC alone and will be especially useful for researchers leveraging this dataset or the complete dataset, which represents one of the longest-running longitudinal metabolomics studies in a population enriched for Alzheimer’s disease (see Data Availability). These insights are essential for guiding metabolite selection in biomarker discovery and longitudinal disease modeling.

## 5. Conclusions

We evaluated the intra-individual longitudinal stability of untargeted plasma and CSF metabolites measured at four time points approximately 2.5 years apart in cognitively unimpaired participants enrolled in the Wisconsin Registry for Alzheimer’s Prevention and Wisconsin Alzheimer’s Disease Research Center studies. Although stability varied widely by metabolite and super-pathway, the majority of metabolites in both fluids demonstrated ‘fair’ longitudinal stability.

By integrating non-parametric ICC estimates with technical variance into a composite score, we identified metabolites that are not only stable over time but also robust to batch effects, an essential consideration for longitudinal and multi-cohort metabolomics studies. Our findings reveal important fluid and pathway-specific differences in reliability, highlight the heightened technical sensitivity of lipids, and underscore the limitations of relying on ICC alone when evaluating longitudinal metabolite stability.

Together, these results provide practical guidance for metabolite selection, study design, and interpretation in longitudinal metabolomics research and contribute to ongoing efforts to improve biomarker discovery and disease modeling in Alzheimer’s disease and related neurodegenerative disorders.

## Figures and Tables

**Figure 1 metabolites-16-00035-f001:**
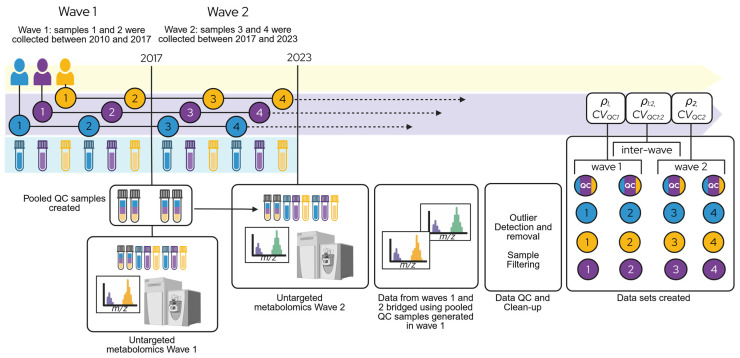
Schematic representation of study workflow. Participant samples were collected and analyzed in two waves. For wave 1, untargeted metabolomics were performed in 2017 on samples collected between 2010 and 2017. A second round of metabolomics was performed in 2023 on samples collected between 2017 and 2023. Metabolomics measurements were bridged using pooled QC samples generated in the first wave (multi-colored tubes/circles). Data sets were compiled following QC and filtering. For each participant, the two most recent visits from wave 1 and the two earliest visits from wave 2 were selected. Visit numbers 1–4 were assigned after filtering, and are indicated in solid colored circles. The inter-wave data set consists of visits 2 (wave 1) and 3 (wave 2). Separate statistical metrics were calculated on each data set, including Rothery’s ICC (*ρ*) and the coefficients of variation of pooled QC samples (*CV_QC_*). Created in BioRender. ROCHA, B. (2025) https://BioRender.com/5tbzxm8 (accessed on 20 December 2025).

**Figure 2 metabolites-16-00035-f002:**
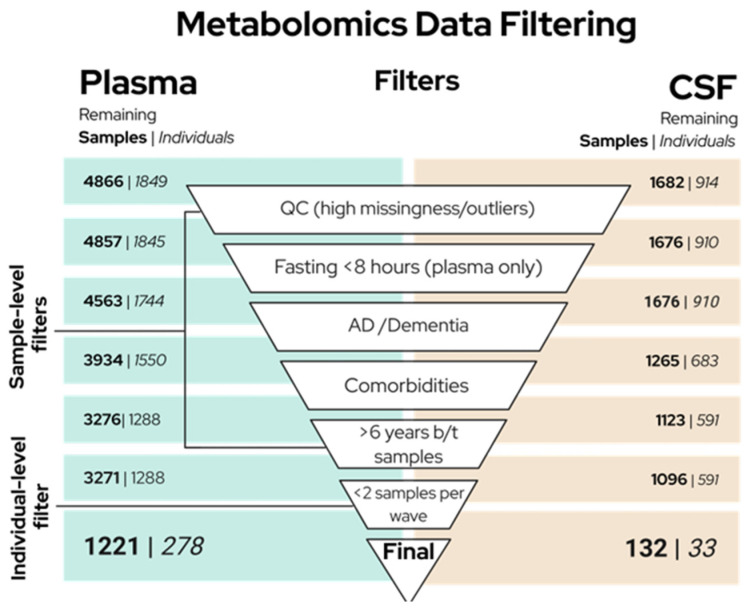
The order in which QC and filtering steps were applied to plasma (left) and CSF (right) data sets is displayed. Brackets and annotations indicate whether each step was applied at the sample or participant level. The number of samples (bolded) and participants (italicized) remaining after each step are indicated. After the final filter was applied, 4 samples from the remaining individuals were selected (see Compiling Data Sets section). The last row reflects the final sample and participant counts. Created in BioRender. ROCHA, B. (2025) https://BioRender.com/wugedzo (accessed on 20 December 2025).

**Figure 3 metabolites-16-00035-f003:**
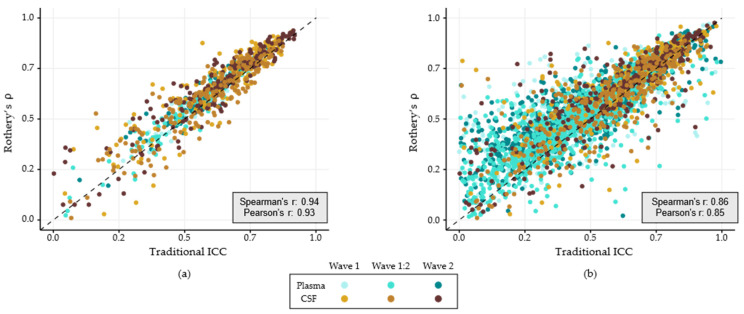
Correlation between traditional intraclass correlation coefficient (ICC; *x*-axis) and Rothery’s *ρ* (*y*-axis) for (**a**) a subset of normally distributed metabolites and (**b**) all metabolites regardless of distribution. Point color indicates data set (categorical), and the dashed black line represents perfect correlation.

**Figure 4 metabolites-16-00035-f004:**
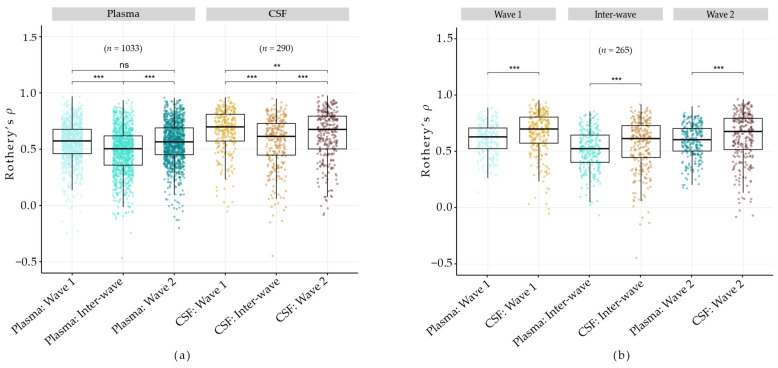
Boxplots of Rothery’s *ρ* (*y*-axis) for all metabolites (**a**) and metabolites detected in both plasma and CSF (**b**). Each point represents an individual metabolite, with point color indicating the data set. The bold line within each box represents the median value for each data set (*x*-axis). Asterisks indicate the degree of significance for pairwise comparisons and were determined using *p*-values from Wilcoxon rank-sum tests (** *p* < 0.01, *** *p* < 0.001, ‘ns’ = not significant).

**Figure 5 metabolites-16-00035-f005:**
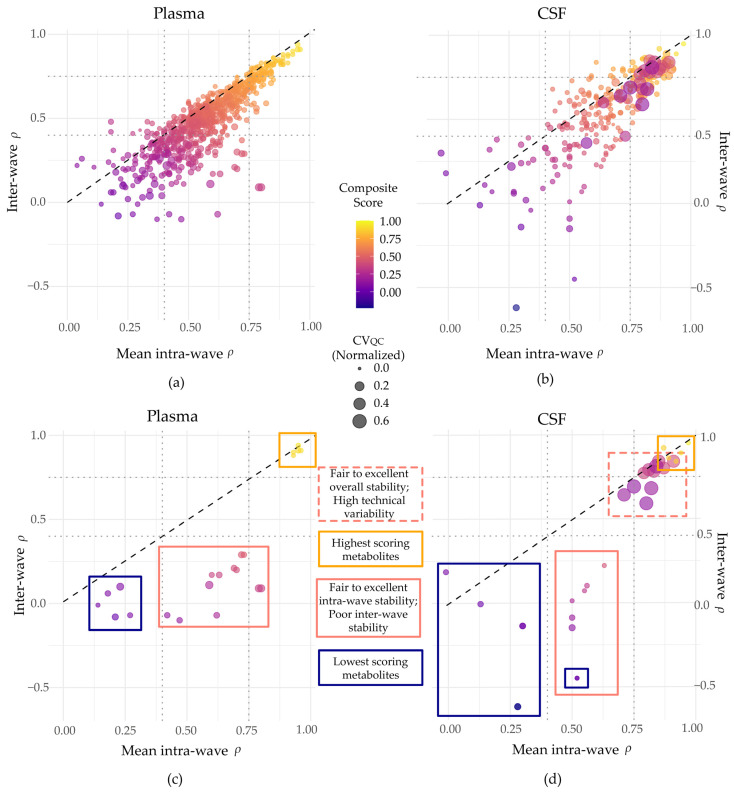
For plasma (**a**,**c**) and CSF (**b**,**d**) metabolites, the mean intra-wave ρ values (*x*-axis) are plotted against the inter-wave ρ values (*y*-axis). Point color represents the composite score for each metabolite (continuous), where yellow indicates ‘high’ scores and dark blue indicates ‘low’ scores. The size of each point represents the average ${\hat{CV}}_{\mathrm{QC}}$ value, with larger points indicating higher technical variance. The top panels show all metabolites, whereas the bottom panels (**c**,**d**) highlight a subset of metabolites for illustrative purposes. In the bottom panels, bounding boxes highlight specific groups including the highest scoring metabolites (yellow boxes), the lowest scoring metabolites (dark blue boxes), as well as select metabolites such as those with ‘fair’ to ‘excellent’ average intra-wave stability (*ρ* > 0.40) but ‘poor’ inter-wave stability (*ρ* ≤ 0.40; solid pink boxes), and those that despite having high inter- and intra-wave stability, have low composite scores due to high technical variance (CSF only; dashed pink box). In CSF, one of the lowest-scoring metabolites displayed ‘fair’ intra-wave stability and, thus, is bounded by both dark blue and pink boxes (**d**).

**Figure 6 metabolites-16-00035-f006:**
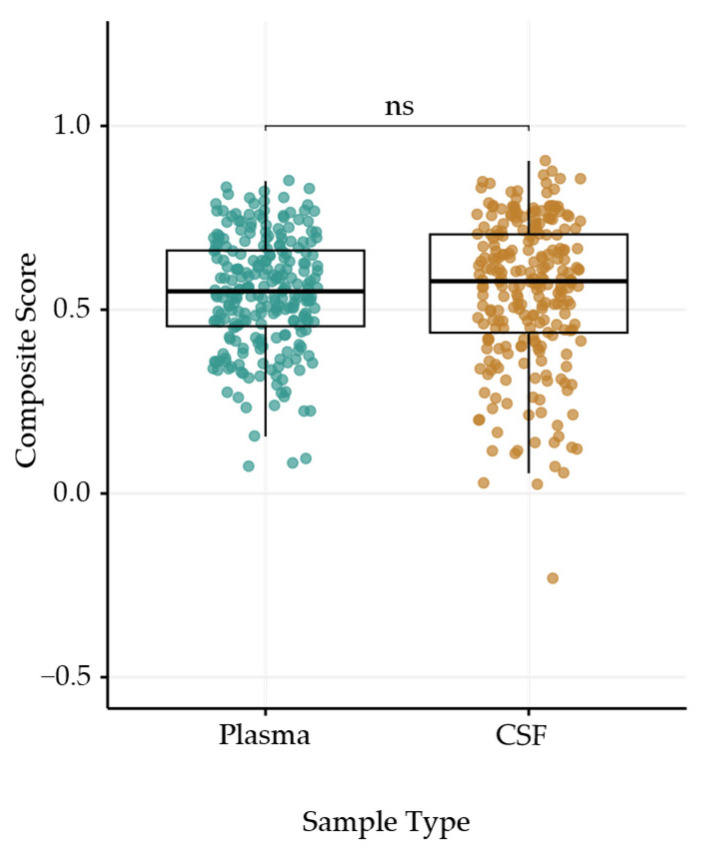
Boxplots showing composite scores among metabolites detected in both plasma (left; teal) and CSF (right; tan). Boxes divide the second and third quartiles, with the bolded line representing the median composite score value. A paired Wilcoxon rank-sum test indicated no significant (ns; *p* > 0.05) difference between composite scores for plasma and CSF.

**Table 1 metabolites-16-00035-t001:** Summary statistics for data included in this analysis.

	Plasma			CSF		
*n* (Samples)	1112			132		
*n* (Participants)	278			33		
% Female	64.4			63.6		
% WRAP ^1^	100			42.4		
	Wave 1	Wave 1:2	Wave 2	Wave 1	Wave 1:2	Wave 2
Visits included	1, 2	2, 3	3, 4	1, 2	2, 3	3, 4
Mean age at baseline	61.2	63.5	66	62.2	65	67.4
Mean time between samples (years)	2.4	2.5	2.9	2.7	2.4	2.3

^1^. For wave 1 (2017), only samples from WRAP participants were used due to ADRC samples being collected using a different anticoagulant.

**Table 2 metabolites-16-00035-t002:** Median Rothery’s *ρ* value and the proportion of metabolites given ‘excellent’ (*ρ* > 0.75), ‘fair’ (0.4 > *ρ* ≤ 0.75), and ‘poor’ (*ρ* ≤ 0.4) stability ratings for plasma and CSF.

	All Metabolites ^1^					
	Plasma (*n* = 1033 Metabolites)			CSF (*n* = 290 Metabolites)		
	Wave 1	Inter-Wave	Wave 2	Wave 1	Inter-Wave	Wave 2
Visits	1:2	2:3	3:4	1:2	2:3	3:4
Median *ρ* (IQR)	0.57 (0.22)	0.50 (0.26)	0.56 (0.24)	0.70 (0.24)	0.61 (0.29)	0.68 (0.29)
% ‘Excellent’ (*n*)	13.4 (138)	8.6 (89)	15.3 (158)	36.6 (106)	20.3 (59)	37.2 (108)
% ‘Fair’ (*n*)	71.7 (741)	60.8 (628)	66.9 (691)	53.4 (155)	57.2 (166)	45.9 (133)
% ‘Poor’ (*n*)	14.9 (154)	30.6 (316)	17.8 (184)	10.0 (29)	22.4 (65)	16.9 (49)
	Common Metabolites ^2^ (*n* = 265 Metabolites)					
Median *ρ* (IQR)	0.63 (0.18)	0.52 (0.24)	0.60 (0.20)	0.70 (0.23)	0.61 (0.29)	0.68 (0.28)
% ‘Excellent’ (*n*)	15.1 (40)	8.3 (22)	15.5 (41)	35.8 (95)	20.0 (53)	37.4 (99)
% ‘Fair’ (*n*)	75.8 (201)	67.9 (180)	72.5 (192)	54.3 (144)	57.7 (153)	46.8 (124)
% ‘Poor’ (*n*)	9.1 (24)	23.8 (63)	23.1 (32)	9.8 (26)	22.3 (59)	15.8 (42)

^1^ All metabolites detected and analyzed in plasma and/or CSF. ^2^ Metabolites that were detected in both plasma and CSF.

**Table 3 metabolites-16-00035-t003:** Plasma and CSF metabolites plotted in Figure 5.

				**Mean Intra-** **Wave *ρ***	**Inter-Wave *ρ***	**Mean** ${CV}_{\mathrm{QC}}$ (Norm.)	**Composite** **Score**
		Metabolite	Super Pathway				
**Highest Scoring**	Plasma	N2-acetyl,N6-methyllysine	Amino Acid	0.95	0.94	0.02	**0.92**
		5alpha-androstan-3alpha,17beta-diol monosulfate (2)	Lipid	0.96	0.91	0.02	**0.92**
		N6,N6-dimethyllysine	Amino Acid	0.93	0.90	0.01	**0.90**
		N2-acetyl,N6,N6-dimethyllysine	Amino Acid	0.95	0.91	0.03	**0.90**
		androstenediol (3alpha, 17alpha) monosulfate (3)	Lipid	0.93	0.88	0.01	**0.90**
	CSF	N6-methyllysine	Amino Acid	0.97	0.95	0.01	**0.95**
		ethylmalonate	Amino Acid	0.94	0.89	0.01	**0.90**
		2-amino-4-cyanobutanoate	Partially Char. Mol	0.87	0.92	0.02	**0.88**
		homocarnosine	Amino Acid	0.92	0.84	0.01	**0.87**
		N-acetylputrescine	Amino Acid	0.89	0.86	0.01	**0.86**
** Lowest Scoring **	Plasma	cytosine	Nucleotide	0.18	0.06	0.04	** 0.08 **
		benzoate	Xenobiotics	0.23	0.10	0.09	** 0.07 **
		maltotriose	Carbohydrate	0.27	−0.07	0.03	** 0.07 **
		iminodiacetate (IDA)	Xenobiotics	0.14	−0.01	0.01	** 0.06 **
		chiro-inositol	Lipid	0.21	−0.08	0.06	** 0.00 **
	CSF	4-chlorobenzoic acid	Xenobiotics	−0.01	0.18	0.03	** 0.06 **
		tartarate	Xenobiotics	0.30	−0.14	0.05	** 0.03 **
		glutamate	Amino Acid	0.52	−0.45	0.01	** 0.03 **
		isocitrate	Energy	0.13	−0.01	0.04	** 0.02 **
		benzoate	Xenobiotics	0.28	−0.62	0.06	** −0.23 **
**Select Metabolites**	Plasma	alpha-ketoglutarate	Energy	0.73	0.29	0.04	** 0.47 **
		bilirubin degradation product, C16H18N2O5 (2) **	Partially Char. Mol	0.72	0.29	0.06	** 0.44 **
		bilirubin (E,E) *	Cofactors and Vitamins	0.70	0.20	0.05	** 0.40 **
		myo-inositol	Lipid	0.69	0.21	0.05	** 0.40 **
		cysteine	Amino Acid	0.60	0.17	0.03	** 0.36 **
		adenosine	Nucleotide	0.63	0.17	0.05	** 0.35 **
		bilirubin degradation product, C17H20N2O5 (2) **	Partially Char. Mol	0.80	0.09	0.10	** 0.34 **
		bilirubin degradation product, C17H20N2O5 (1) **	Partially Char. Mol	0.79	0.09	0.10	** 0.34 **
		succinate	Energy	0.59	0.11	0.11	** 0.24 **
		N-acetylasparagine	Amino Acid	0.62	−0.07	0.05	** 0.22 **
		1-methylguanidine	Amino Acid	0.47	−0.10	0.04	** 0.14 **
		cysteine sulfinic acid	Amino Acid	0.42	−0.07	0.04	** 0.14 **
	CSF	N-acetylglutamine	Amino Acid	0.63	0.22	0.01	** 0.42 **
		1-stearoyl-2-arachidonoyl-GPC (18:0/20:4)	Lipid	0.91	0.84	0.47	** 0.41 **
		1-palmitoyl-2-docosahexaenoyl-GPC (16:0/22:6)	Lipid	0.87	0.80	0.46	** 0.38 **
		sphingomyelin (d18:2/16:0, d18:1/16:1) *	Lipid	0.79	0.77	0.41	** 0.37 **
		1-palmitoyl-2-arachidonoyl-GPC (16:0/20:4n6)	Lipid	0.85	0.84	0.50	** 0.34 **
		sphingomyelin (d18:1/24:1, d18:2/24:0) *	Lipid	0.83	0.78	0.46	** 0.34 **
		sphingomyelin (d18:2/24:1, d18:1/24:2) *	Lipid	0.81	0.79	0.48	** 0.32 **
		methionine sulfoxide	Amino Acid	0.56	0.10	0.02	** 0.31 **
		uridine	Nucleotide	0.55	0.07	0.01	** 0.30 **
		malate	Energy	0.50	0.01	0.01	** 0.24 **
		1-palmitoyl-2-linoleoyl-GPC (16:0/18:2)	Lipid	0.84	0.81	0.62	** 0.20 **
		1-stearoyl-GPC (18:0)	Lipid	0.82	0.68	0.58	** 0.17 **
		sphingomyelin (d18:1/22:1, d18:2/22:0, d16:1/24:1) *	Lipid	0.71	0.64	0.51	** 0.16 **
		maleate	Lipid	0.50	−0.09	0.04	** 0.16 **
		behenoyl sphingomyelin (d18:1/22:0) *	Lipid	0.75	0.69	0.58	** 0.14 **
		1-stearoyl-2-linoleoyl-GPC (18:0/18:2) *	Lipid	0.80	0.59	0.56	** 0.14 **

Plasma and CSF metabolites plotted in Figure 5, including those with the highest composite scores, the lowest composite scores, and select metabolites. For each metabolite, the super pathway is indicated, as well as the average intra-wave *ρ*, inter-wave *ρ*, the average normalized *CV_QC_* values (calculated as the average of intra-wave and inter-wave ${\hat{CV}}_{\mathrm{QC}}$ values), and the composite scores. The colored bar on the left corresponds to the color of the bounding boxes displayed in Figure 5c,d. The color of each composite score value corresponds to point color of each metabolite as displayed in Figure 5 where yellow indicates highest composite scores, and dark blue indicates lowest composite scores. Metabolite names ending in * or ** are compounds that have not been confirmed based on the Metabolomics Standards Initiative Tier 1 identification system, where * indicates high confidence, and ** indicates reasonable confidence in compound identity.

## Data Availability

Scientists interested in accessing the data can submit resource requests for the Wisconsin ADRC and WRAP data through the following website: https://www.adrc.wisc.edu/apply-resources (accessed on 20 December 2025).

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
