# Peer review of "Longitudinal, Intra-Individual Stability of Untargeted Plasma and Cerebrospinal Fluid Metabolites"

_metabolites, 2025, doi:10.3390/metabo16010035_

Round 1

Reviewer 1 Report

Comments and Suggestions for Authors

The article “Longitudinal, intra-individual stability of untargeted plasma and cerebrospinal fluid metabolites” provides information on the longitudinal and technical stability of untargeted plasma and cerebrospinal fluid (CSF) metabolites, which allows identification of metabolites suitable for long-term epidemiological studies and offers experimental design and analytical strategies allowing the combination of data across various cohorts and analytical batches. The study presents interesting results, has a perfect design, and should be published. Points for minor corrections presented below, hopefully, will improve the quality of presentation of this interesting study:

Minor changes suggested:

  1. Fig. 4 Signs of the X-axis are hardly visible; font size should be increased.
  2. There is no mention of Table 3 in the results section; however, it is mentioned in the Discussion. All tables and figures should be sequentially mentioned in the results section.
  3. Indeed, based on multiple studies, the common belief is that CSF metabolites are more stable than plasma metabolites. The authors of the study detected the opposite pattern. I think these findings deserve the extension of the discussion and suggestions about the reasons for this finding.
  4. Even though this information can be obtained from the data presented, I would also suggest summarizing in the discussion at least 5 top metabolites in CSF and serum characterized with the highest longitudinal and technical stability and suggesting the reasons for the stability of these groups of compounds based on available information.

Author Response

Thank you very much for your thoughtful comments and suggestions. They are well received. 

Comment 1: 4 Signs of the X-axis are hardly visible; font size should be increased.

Response 1: Suggestion noted, font size adjusted accordingly

Comment 2: There is no mention of Table 3 in the results section; however, it is mentioned in the Discussion. All tables and figures should be sequentially mentioned in the results section.

Response 2: Thank you for catching this discrepancy. Table numbers were updated and all tables are now mentioned in the results section.

Comment 3: Indeed, based on multiple studies, the common belief is that CSF metabolites are more stable than plasma metabolites. The authors of the study detected the opposite pattern. I think these findings deserve the extension of the discussion and suggestions about the reasons for this finding.

Response 3: This comment is greatly appreciated as you highlighted a finding that has been perplexing me. I consulted my co-authors as well as personnel from our clinical sites who have more historical knowledge about our study design and sample handling. As a result, it came to light that the sample storage protocols for CSF were not standardized across sites until December 2018. Prior to this, CSF samples were stored in a variety of tube types. Given that lipids can be especially sensitive to tube material, it is very likely that this difference represents a significant source of technical variation, particularly for visits 1 and 2 which took place before 2018. This is evidenced by the elevated CV values for the lipid super pathway mentioned in the results section (3.4. Stability by pathway).

Per your comments and this new information, I have updated the methods section to report changes made to our CSF sample storage protocols, and expanded the discussion section to include differences in sample tubes as a likely contributor to the elevated CV values observed in our CSF samples. Since we were limited by a retroactive study design, we did not have control over much of the pre-analytical study design. Although unfortunate, this is a common occurrence in longitudinal studies. I believe our results are informative and highlight the importance of careful planning in the pre-analytical stage to reduce sources of technical variance, and when not possible, assessing stability using methods like our composite score that measure and weigh technical variance and batch effects more explicitly.

Comment 4: Even though this information can be obtained from the data presented, I would also suggest summarizing in the discussion at least 5 top metabolites in CSF and serum characterized with the highest longitudinal and technical stability and suggesting the reasons for the stability of these groups of compounds based on available information.

Response 4: In response to this comment, I have added a short passage to the discussion section that addresses the top scoring metabolites for both fluids. Rather than name each metabolite explicitly, I comment on super pathways represented including amino acids which predominate the highest scoring metabolites in both plasma and CSF, and lipids, which are represented among the highest scoring plasma metabolites, but not CSF – likely for reasons described above.

Reviewer 2 Report

Comments and Suggestions for Authors

The manuscript evaluates how stable (repeatable) untargeted LC-MS metabolite measurements are over multi-year longitudinal sampling and across separate analytical “waves” in two Alzheimer’s-related cohorts (WRAP and Wisconsin ADRC), using paired plasma and CSF samples from generally healthy participants.

Using four timepoints per person (~2.5 years apart), with two visits analyzed in 2017 and two analyzed in 2023, the authors quantify within-wave (intra-wave) vs between-wave (inter-wave) stability via Rothery’s intraclass correlation coefficients (ICCs), alongside pooled QC variability to capture technical noise. They then classify metabolites by stability thresholds and create a composite score that prioritizes metabolites that are both longitudinally consistent and less sensitive to batch/wave effects.

Key findings are that overall median stability is “fair” in both plasma and CSF, but interwave stability is substantially lower than intra-wave stability, even with pooled QC bridging—implying technical differences across analytical waves meaningfully degrade comparability. The authors identify subsets of highly stable metabolites and note pathway-level differences in stability intended to guide metabolite selection and study design for long-term epidemiologic metabolomics and for combining data across batches/cohorts.

Overall, the study appears scientifically rigorous and thoughtfully designed, and the manuscript is written with a coherent narrative from motivation through interpretation. That said, I noted a few minor grammatical and stylistic issues below that, if addressed, would further improve clarity and readability.

I suggest to add a couple of sentences or a short paragraph in the introduction detailing the broad importance of CSF metabolites, and the specific relevance of CSF metabolites to studies involving AD.

Section 2.5: Would appreciate more details on instrument analysis. Was biphasic extraction used? Were both polar and nonpolar metabolites analyzed? Was polar phase analyzed with reversed-phase or HILIC chromatography? In positive, negative or switching modes?

Line 171: Be careful of the use of “UPLC” as opposed to “UHPLC”. UHPLC is a broad term that means Ultra-High Performance LC and can be applied to any appropriate LC. UPLC is a trademark of Waters and should only be applied to their specific line of UHPLCs. I’m not sure what was used by Metabolon; Figure 1 depicts a Thermo instrument. If not Waters, or if it is unknown, please change UPLC to UHPLC.

Figure 5: Beautiful figure. I think legibility could be slightly improved by including “Plasma” and “CSF” labels above (c) and (d). Also consider including y-axis label and title on (b) and (d).

Figure 5 (d): Why does one point appear to be boxed in by both blue and pink box?

Author Response

Your review and comments are much appreciated. 

Comment 1: I suggest to add a couple of sentences or a short paragraph in the introduction detailing the broad importance of CSF metabolites, and the specific relevance of CSF metabolites to studies involving AD.

Response 1: Thank you for this comment. I have added the requested information to the introduction.

Comment 2: Section 2.5: Would appreciate more details on instrument analysis. Was biphasic extraction used? Were both polar and nonpolar metabolites analyzed? Was polar phase analyzed with reversed-phase or HILIC chromatography? In positive, negative or switching modes?

Response 2: Given that a key focus of our manuscript focuses on sources of technical variance, your point is well taken. I have added substantial detail to the methods section that answers your specific questions and more.

Comment 3: Line 171: Be careful of the use of “UPLC” as opposed to “UHPLC”. UHPLC is a broad term that means Ultra-High Performance LC and can be applied to any appropriate LC. UPLC is a trademark of Waters and should only be applied to their specific line of UHPLCs. I’m not sure what was used by Metabolon; Figure 1 depicts a Thermo instrument. If not Waters, or if it is unknown, please change UPLC to UHPLC.

Response 3: Thank you for pointing this out. I was not aware of this distinction, but confirmed that Metabolon does indeed use a Waters instrument. I retained the use of UPLC but ensured consistency in its use.            

Comment 4: Figure 5: Beautiful figure. I think legibility could be slightly improved by including “Plasma” and “CSF” labels above (c) and (d). Also consider including y-axis label and title on (b) and (d).

Response 4: I have taken your suggestions and added the labels and titles.

Comment 5: Figure 5 (d): Why does one point appear to be boxed in by both blue and pink box?

Response 5: To ensure clarity, I have added the following sentence to clarify: “In CSF, one of the lowest scoring metabolites displayed fair intra-wave stability and thus, is bounded by both dark blue and pink boxes (d).”